# Memory-Efficient Backpropagation Through Time

**Audrūnas Gruslys**
Google DeepMind
audrunas@google.com

**Rémi Munos**
Google DeepMind
munos@google.com

**Ivo Danihelka**
Google DeepMind
danihelka@google.com

**Marc Lanctot**
Google DeepMind
lanctot@google.com

**Alex Graves**
Google DeepMind
gravesa@google.com

## Abstract

We propose a novel approach to reduce memory consumption of the backpropagation through time (BPTT) algorithm when training recurrent neural networks (RNNs). Our approach uses dynamic programming to balance a trade-off between caching of intermediate results and recomputation. The algorithm is capable of tightly fitting within almost any user-set memory budget while finding an optimal execution policy minimizing the computational cost. Computational devices have limited memory capacity and maximizing a computational performance given a fixed memory budget is a practical use-case. We provide asymptotic computational upper bounds for various regimes. The algorithm is particularly effective for long sequences. For sequences of length 1000, our algorithm saves 95% of memory usage while using only one third more time per iteration than the standard BPTT.

## 1  Introduction

Recurrent neural networks (RNNs) are artificial neural networks where connections between units can form cycles. They are often used for sequence mapping problems, as they can propagate hidden state information from early parts of the sequence back to later points. LSTM [9] in particular is an RNN architecture that has excelled in sequence generation [3, 13, 4], speech recognition [5] and reinforcement learning [12, 10] settings. Other successful RNN architectures include the differentiable neural computer (DNC) [6], DRAW network [8], and Neural Transducers [7].

Backpropagation Through Time algorithm (BPTT) [11, 14] is typically used to obtain gradients during training. One important problem is the large memory consumption required by the BPTT. This is especially troublesome when using Graphics Processing Units (GPUs) due to the limitations of GPU memory.

Memory budget is typically known in advance. Our algorithm balances the tradeoff between memorization and recomputation by finding an optimal memory usage policy which minimizes the total computational cost for any fixed memory budget. The algorithm exploits the fact that the same memory slots may be reused multiple times. The idea to use dynamic programming to find a provably optimal policy is the main contribution of this paper.

Our approach is largely architecture agnostic and works with most recurrent neural networks. Being able to fit within limited memory devices such as GPUs will typically compensate for any increase in computational cost.

## 2  Background and related work

In this section, we describe the key terms and relevant previous work for memory-saving in RNNs.

**Definition 1.** *An **RNN core** is a feed-forward neural network which is cloned (unfolded in time) repeatedly, where each clone represents a particular time point in the recurrence.*

For example, if an RNN has a single hidden layer whose outputs feed back into the same hidden layer, then for a sequence length of $t$ the unfolded network is feed-forward and contains $t$ RNN cores.

**Definition 2.** *The **hidden state** of the recurrent network is the part of the output of the RNN core which is passed into the next RNN core as an input.*

In addition to the initial hidden state, there exists a single hidden state per time step once the network is unfolded.

**Definition 3.** *The **internal state** of the RNN core for a given time-point is all the necessary information required to backpropagate gradients over that time step once an input vector, a gradient with respect to the output vector, and a gradient with respect to the output hidden state is supplied. We define it to also include an output hidden state.*

An internal state can be (re)evaluated by executing a single forward operation taking the previous hidden state and the respective entry of an input sequence as an input. For most network architectures, the internal state of the RNN core will include a hidden input state, as this is normally required to evaluate gradients. This particular choice of the definition will be useful later in the paper.

**Definition 4.** *A **memory slot** is a unit of memory which is capable of storing a single hidden state or a single internal state (depending on the context).*

## 2.1 Backpropagation through Time

Backpropagation through Time (BPTT) [11, 14] is one of the commonly used techniques to train recurrent networks. BPTT "unfolds" the neural network in time by creating several copies of the recurrent units which can then be treated like a (deep) feed-forward network with tied weights. Once this is done, a standard forward-propagation technique can be used to evaluate network fitness over the whole sequence of inputs, while a standard backpropagation algorithm can be used to evaluate partial derivatives of the loss criteria with respect to all network parameters. This approach, while being computationally efficient is also fairly intensive in memory usage. This is because the standard version of the algorithm effectively requires storing internal states of the unfolded network core at every time-step in order to be able to evaluate correct partial derivatives.

## 2.2 Trading memory for computation time

The general idea of trading computation time and memory consumption in general computation graphs has been investigated in the automatic differentiation community [2]. Recently, the rise of deep architectures and recurrent networks has increased interest in a less general case where the graph of forward computation is a chain and gradients have to be chained in a reverse order. This simplification leads to relatively simple memory-saving strategies and heuristics. In the context of BPTT, instead of storing hidden network states, some of the intermediate results can be recomputed on demand by executing an extra forward operation.

Chen et. al. proposed subdividing the sequence of size $t$ into $\sqrt{t}$ equal parts and memorizing only hidden states between the subsequences and all internal states within each segment [1]. This uses $O(\sqrt{t})$ memory at the cost of making an additional forward pass on average, as once the errors are backpropagated through the right-side of the sequence, the second-last subsequence has to be restored by repeating a number of forward operations. We refer to this as Chen's $\sqrt{t}$ algorithm.

The authors also suggest applying the same technique recursively several times by sub-dividing the sequence into $k$ equal parts and terminating the recursion once the subsequence length becomes less than $k$. The authors have established that this would lead to memory consumption of $O(k \log_{k+1}(t))$ and computational complexity of $O(t \log_k(t))$. This algorithm has a minimum possible memory usage of $\log_2(t)$ in the case when $k = 1$. We refer to this as Chen's *recursive* algorithm.

## 3 Memory-efficient backpropagation through time

We first discuss two simple examples: when memory is very scarce, and when it is somewhat limited.

When memory is very scarce, it is straightforward to design a simple but computationally inefficient algorithm for backpropagation of errors on RNNs which only uses a constant amount of memory. Every time when the state of the network at time $t$ has to be restored, the algorithm would simply re-evaluate the state by forward-propagating inputs starting from the beginning until time $t$. As backpropagation happens in the reverse temporal order, results from the previous forward steps can not be reused (as there is no memory to store them). This would require repeating $t$ forward steps before backpropagating gradients one step backwards (we only remember inputs and the initial state). This would produce an algorithm requiring $t(t+1)/2$ forward passes to backpropagate errors over $t$ time steps. The algorithm would be $O(1)$ in space and $O(t^2)$ in time.

When the memory is somewhat limited (but not very scarce) we may store only hidden RNN states at all time points. When errors have to be backpropagated from time $t$ to $t-1$, an internal RNN core state can be re-evaluated by executing another forward operation taking the previous hidden state as an input. The backward operation can follow immediately. This approach can lead to fairly significant memory savings, as typically the recurrent network hidden state is much smaller than an internal state of the network core itself. On the other hand this leads to another forward operation being executed during the backpropagation stage.

### 3.1 Backpropagation though time with selective hidden state memorization (BPTT-HSM)

The idea behind the proposed algorithm is to compromise between two previous extremes. Suppose that we want to forward and backpropagate a sequence of length $t$, but we are only able to store $m$ hidden states in memory at any given time. We may reuse the same memory slots to store different hidden states during backpropagation. Also, suppose that we have a single RNN core available for the purposes of intermediate calculations which is able to store a single internal state. Define $C(t, m)$ as a computational cost of backpropagation measured in terms of how many forward-operations one has to make in total during forward and backpropagation steps combined when following an optimal memory usage policy minimizing the computational cost. One can easily set the boundary conditions: $C(t, 1) = \frac{1}{2}t(t+1)$ is the cost of the minimal memory approach, while $C(t, m) = 2t - 1$ for all $m \geq t$ when memory is plentiful (as shown in Fig. 3 a). Our approach is illustrated in Figure 1. Once we start forward-propagating steps at time $t = t_0$, at any given point $y > t_0$ we can choose to put the current hidden state into memory (step 1). This step has the cost of $y$ forward operations. States will be read in the reverse order in which they were written: this allows the algorithm to store states in a stack. Once the state is put into memory at time $y = D(t, m)$, we can reduce the problem into two parts by using a divide-and-conquer approach: running the same algorithm on the $t > y$ side of the sequence while using $m - 1$ of the remaining memory slots at the cost of $C(t - y, m - 1)$ (step 2), and then reusing $m$ memory slots when backpropagating on the $t \leq y$ side at the cost of $C(y, m)$ (step 3). We use a full size $m$ memory capacity when performing step 3 because we could release the hidden state $y$ immediately after finishing step 2.

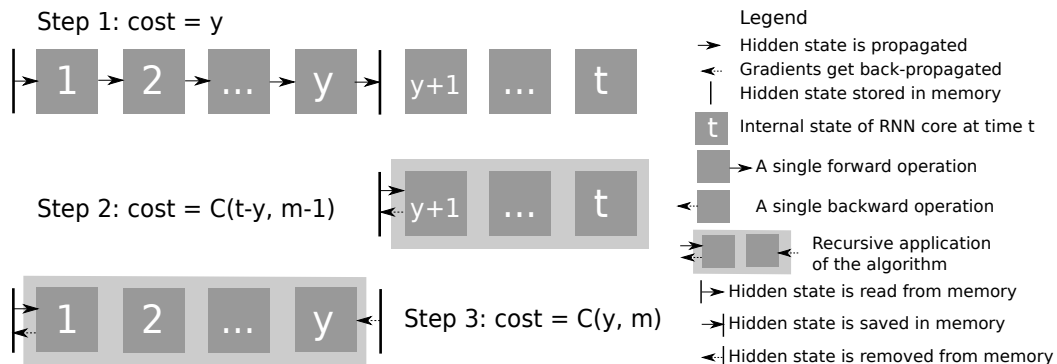

Figure 1: The proposed divide-and-conquer approach.

The base case for the recurrent algorithm is simply a sequence of length $t = 1$ when forward and backward propagation may be done trivially on a single available RNN network core. This step has the cost $C(1, m) = 1$.

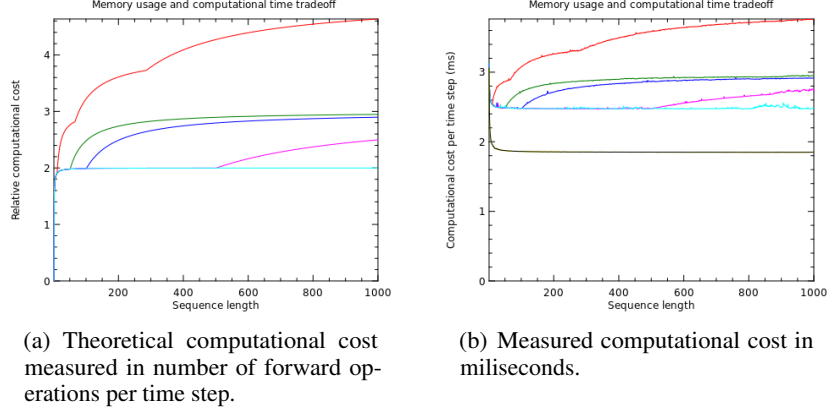

(a) Theoretical computational cost measured in number of forward operations per time step.

(b) Measured computational cost in miliseconds.

Figure 2: Computational cost per time-step when the algorithm is allowed to remember 10 (red), 50 (green), 100 (blue), 500 (violet), 1000 (cyan) hidden states. The grey line shows the performance of standard BPTT without memory constraints; (b) also includes a large constant value caused by a single backwards step per time step which was excluded from the theoretical computation, which value makes a relative performance loss much less severe in practice than in theory.

Having established the protocol we may find an optimal policy $D(t, m)$. Define the cost of choosing the first state to be pushed at position $y$ and later following the optimal policy as:

$$Q(t, m, y) = y + C(t - y, m - 1) + C(y, m) \tag{1}$$

$$C(t, m) = Q(t, m, D(t, m)) \tag{2}$$

$$D(t, m) = \operatorname*{argmin}_{1 \leq y < t} Q(t, m, y) \tag{3}$$

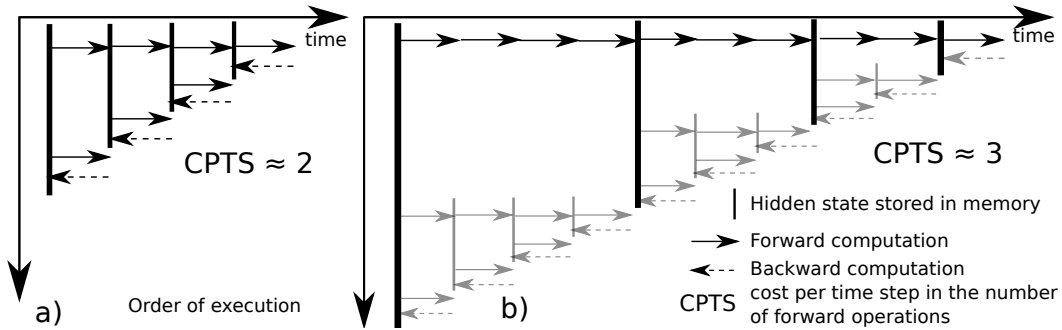

Figure 3: Illustration of the optimal policy for $m = 4$ and a) $t = 4$ and b) $t = 10$. Logical sequence time goes from left to right, while execution happens from top to the bottom.

Equations can be solved exactly by using dynamic programming subject to the boundary conditions established previously (*e.g.* as in Figure 2(a)). $D(t, m)$ will determine the optimal policy to follow. Pseudocode is given in the supplementary material. Figure 3 illustrates an optimal policy found for two simple cases.

## 3.2 Backpropagation though time with selective internal state memorization (BPTT-ISM)

Saving internal RNN core states instead of hidden RNN states would allow us to save a single forward operation during backpropagation in every divide-and-conquer step, but at a higher memory cost. Suppose we have a memory capacity capable of saving exactly $m$ internal RNN states. First, we need to modify the boundary conditions: $C(t, 1) = \frac{1}{2}t(t + 1)$ is a cost reflecting the minimal memory approach, while $C(t, m) = t$ for all $m \geq t$ when memory is plentiful (equivalent to standard BPTT).

As previously, $C(t, m)$ is defined to be the computational cost for combined forward and backward propagations over a sequence of length $t$ with memory allowance $m$ while following an optimal memory usage policy. As before, the cost is measured in terms of the amount of total forward steps made, because the number of backwards steps is constant. Similarly to BPTT-HSM, the process can be divided into parts using divide-and-conquer approach (Fig 4). For any values of $t$ and $m$ position of the first memorization $y = D(t, m)$ is evaluated. $y$ forward operations are executed and an internal RNN core state is placed into memory. This step has the cost of $y$ forward operations (Step 1 in Figure 4). As the internal state also contains an output hidden state, the same algorithm can be recurrently run on the high-time (right) side of the sequence while having one less memory slot available (Step 2 in Figure 4). This step has the cost of $C(t - y, m - 1)$ forward operations. Once gradients are backpropagated through the right side of the sequence, backpropagation can be done over the stored RNN core (Step 3 in Figure 4). This step has no additional cost as it involves no more forward operations. The memory slot can now be released leaving $m$ memory available. Finally, the same algorithm is run on the left-side of the sequence (Step 4 in Figure 4). This final step has the cost of $C(y - 1, m)$ forward operations. Summing the costs gives us the following equation:

$$Q(t, m, y) = y + C(y - 1, m) + C(t - y, m - 1) \qquad (4)$$

Recursion has a single base case: backpropagation over an empty sequence is a nil operation which has no computational cost making $C(0, m) = 0$.

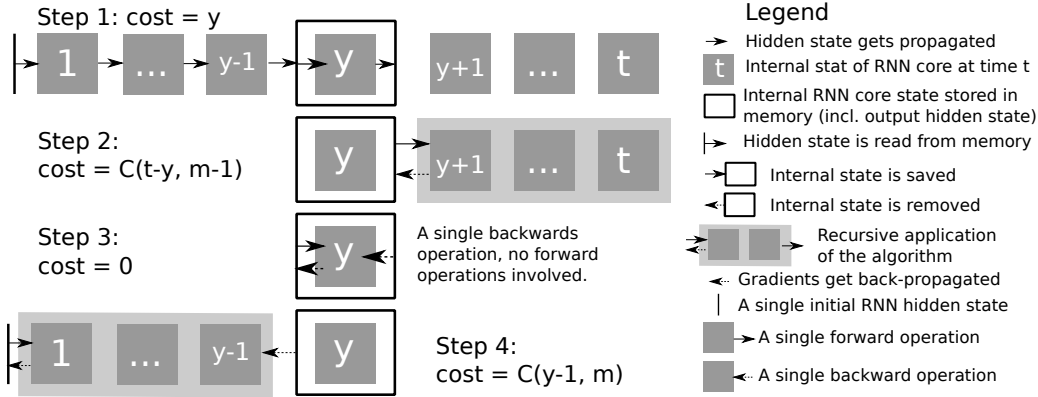

Figure 4: Illustration of the divide-and-conquer approach used by BPTT-ISM.

Compared to the previous section (20) stays the same while (19) is minimized over $1 \leq y \leq t$ instead of $1 \leq y < t$. This is because it is meaningful to remember the last internal state while there was no reason to remember the last hidden state. A numerical solution of $C(t, m)$ for several different memory capacities is shown in Figure 5(a).

$$D(t, m) = \underset{1 \leq y \leq t}{\operatorname{argmin}} Q(t, m, y) \qquad (5)$$

As seen in Figure 5(a), our methodology saves 95% of memory for sequences of 1000 (excluding input vectors) while using only 33% more time per training-iteration than the standard BPTT (assuming a single backward step being twice as expensive as a forward step).

## 3.3 Backpropagation though time with mixed state memorization (BPTT-MSM)

It is possible to derive an even more general model by combining both approaches as described in Sections 3.1 and 3.2. Suppose we have a total memory capacity $m$ measured in terms of how much a single hidden states can be remembered. Also suppose that storing an internal RNN core state takes $\alpha$ times more memory where $\alpha \geq 2$ is some integer number. We will choose between saving a single hidden state while using a single memory unit and storing an internal RNN core state by using $\alpha$ times more memory. The benefit of storing an internal RNN core state is that we will be able to save a single forward operation during backpropagation.

Define $C(t, m)$ as a computational cost in terms of a total amount of forward operations when running an optimal strategy. We use the following boundary conditions: $C(t, 1) = \frac{1}{2}t(t + 1)$ as a

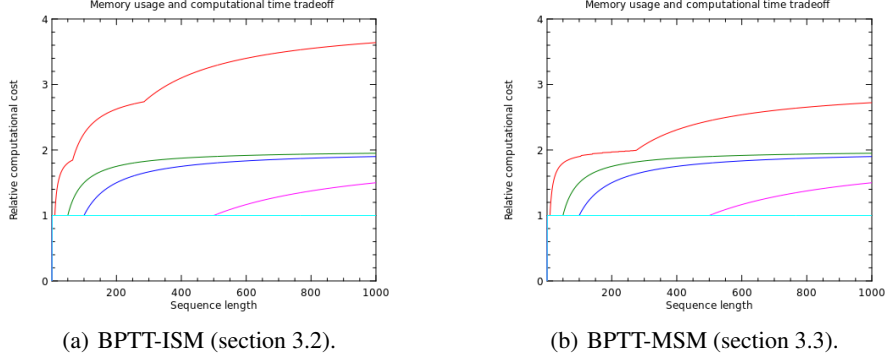

(a) BPTT-ISM (section 3.2).    (b) BPTT-MSM (section 3.3).

Figure 5: Comparison of two backpropagation algorithms in terms of theoretical costs. Different lines show the number of forward operations per time-step when the memory capacity is limited to 10 (red), 50 (green), 100 (blue), 500 (violet), 1000 (cyan) internal RNN core states. Please note that the units of memory measurement are different than in Figure 2(a) (size of an internal state vs size of a hidden state). It was assumed that the size of an internal core state is $\alpha = 5$ times larger than the size of a hidden state. The value of $\alpha$ influences only the right plot. All costs shown on the right plot should be less than the respective costs shown on the left plot for any value of $\alpha$.

cost reflecting the minimal memory approach, while $C(t, m) = t$ for all $m \geq \alpha t$ when memory is plentiful and $C(t - y, m) = \infty$ for all $m \leq 0$ and $C(0, m) = 0$ for notational convenience. We use a similar divide-and-conquer approach to the one used in previous sections.

Define $Q_1(t, m, y)$ as the computational cost if we choose to firstly remember a hidden state at position $y$ and thereafter follow an optimal policy (identical to ( 18)):

$$Q_1(t, m, y) = y + C(y, m) + C(t - y, m - 1) \tag{6}$$

Similarly, define $Q_2(t, m, y)$ as the computational cost if we choose to firstly remember an internal state at position $y$ and thereafter follow an optimal policy (similar to ( 4) except that now the internal state takes $\alpha$ memory units):

$$Q_2(t, m, y) = y + C(y - 1, m) + C(t - y, m - \alpha) \tag{7}$$

Define $D_1$ as an optimal position of the next push assuming that the next state to be pushed is a hidden state and define $D_2$ as an optimal position if the next push is an internal core state. Note that $D_2$ has a different range over which it is minimized, for the same reasons as in equation 5:

$$D_1(t, m) = \operatorname*{argmin}_{1 \leq y < t} Q_1(t, m, y) \qquad D_2(t, m) = \operatorname*{argmin}_{1 \leq y \leq t} Q_2(t, m, y) \tag{8}$$

Also define $C_i(t, m) = Q_i(t, m, D(t, m))$ and finally:

$$C(t, m) = \min_i C_i(t, m) \qquad H(t, m) = \operatorname*{argmin}_i C_i(t, m) \tag{9}$$

We can solve the above equations by using simple dynamic programming. $H(t, m)$ will indicate whether the next state to be pushed into memory in a hidden state or an internal state, while the respective values if $D_1(t, m)$ and $D_2(t, m)$ will indicate the position of the next push.

## 3.4   Removing double hidden-state memorization

Definition 3 of internal RNN core state would typically require for a hidden input state to be included for each memorization. This may lead to the duplication of information. For example, when an optimal strategy is to remember a few internal RNN core states in sequence, a memorized hidden output of one would be equal to a memorized hidden input for the other one (see Definition 3).

Every time we want to push an internal RNN core state onto the stack and a previous internal state is already there, we may omit pushing the input hidden state. Recall that an internal core RNN state when an input hidden state is otherwise not known is $\alpha$ times larger than a hidden state. Define $\beta \leq \alpha$ as the space required to memorize the internal core state when an input hidden state is known. A

relationship between $\alpha$ and $\beta$ is application-specific, but in many circumstances $\alpha = \beta + 1$. We only have to modify (7) to reflect this optimization:

$$Q_2(t, m, y) = y + C(y - 1, m) + C(t - y, m - \mathbb{1}_{y>1}\alpha - \mathbb{1}_{y=1}\beta) \qquad (10)$$

$\mathbb{1}$ is an indicator function. Equations for $H(t, m)$, $D_i(t, m)$ and $C(t, m)$ are identical to (8) and (9).

### 3.5 Analytical upper bound for BPTT-HSM

We have established a theoretical upper bound for BPTT-HSM algorithm as $C(t, m) \leq mt^{1+\frac{1}{m}}$. As the bound is not tight for short sequences, it was also numerically verified that $C(t, m) < 4t^{1+\frac{1}{m}}$ for $t < 10^5$ and $m < 10^3$, or less than $3t^{1+\frac{1}{m}}$ if the initial forward pass is excluded. In addition to that, we have established a different bound in the regime where $t < \frac{m^m}{m!}$. For any integer value $a$ and for all $t < \frac{m^a}{a!}$ the computational cost is bounded by $C(t, m) \leq (a + 1)t$. The proofs are given in the supplementary material. Please refer to supplementary material for discussion on the upper bounds for BPTT-MSM and BPTT-ISM.

### 3.6 Comparison of the three different strategies

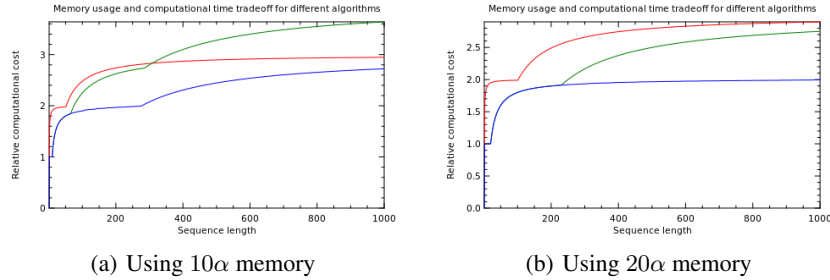

(a) Using $10\alpha$ memory        (b) Using $20\alpha$ memory

Figure 6: Comparison of three strategies in the case when a size of an internal RNN core state is $\alpha = 5$ times larger than that of the hidden state, and the total memory capacity allows us remember either 10 internal RNN states, or 50 hidden states or any arbitrary mixture of those in the left plot and (20, 100) respectively in the right plot. The red curve illustrates BPTT-HSM, the green curve - BPTT-ISM and the blue curve - BPTT-MSM. Please note that for large sequence lengths the red curve out-performs the green one, and the blue curve outperforms the other two.

Computational costs for each previously described strategy and the results are shown in Figure 6. BPTT-MSM outperforms both BPTT-ISM and BPTT-HSM. This is unsurprising, because the search space in that case is a superset of both strategy spaces, while the algorothm finds an optimal strategy within that space. Also, for a fixed memory capacity, the strategy memorizing only hidden states outperforms a strategy memorizing internal RNN core states for long sequences, while the latter outperforms the former for relatively short sequences.

## 4 Discussion

We used an LSTM mapping 256 inputs to 256 with a batch size of 64 and measured execution time for a single gradient descent step (forward and backward operation combined) as a function of sequence length (Figure 2(b)). Please note that measured computational time also includes the time taken by backward operations at each time-step which dynamic programming equations did not take into the account. A single backward operation is usually twice as expensive than a forward operation, because it involves evaluating gradients both with respect to input data and internal parameters. Still, as the number of backward operations is constant it has no impact on the optimal strategy.

### 4.1 Optimality

The dynamic program finds the optimal computational strategy by construction, subject to memory constraints and a fairly general model that we impose. As both strategies proposed by [1] are

consistent with all the assumptions that we have made in section 3.4 when applied to RNNs, BPTT-MSM is guaranteed to perform at least as well under any memory budget and any sequence length. This is because strategies proposed by [1] can be expressed by providing a (potentially suboptimal) policy $D_i(t, m), H(t, m)$ subject to the same equations for $Q_i(t, m)$.

## 4.2 Numerical comparison with Chen's $\sqrt{t}$ algorithm

Chen's $\sqrt{t}$ algorithm requires to remember $\sqrt{t}$ hidden states and $\sqrt{t}$ internal RNN states (excluding input hidden states), while the recursive approach requires to remember at least $\log_2 t$ hidden states. In other words, the model does not allow for a fine-grained control over memory usage and rather saves some memory. In the meantime our proposed BPTT-MSM can fit within almost arbitrary constant memory constraints, and this is the main advantage of our algorithm.

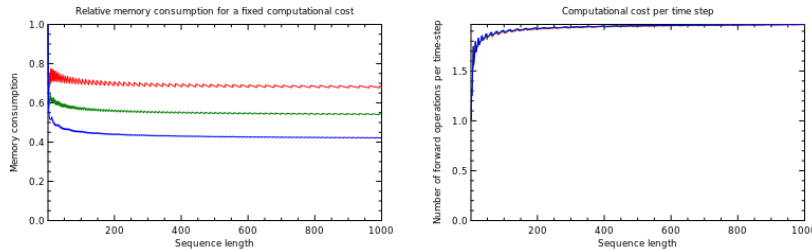

Figure 7: Left: memory consumption divided by $\sqrt{t}(1 + \beta)$ for a fixed computational cost $C = 2$. Right: computational cost per time-step for a fixed memory consumption of $\sqrt{t}(1 + \beta)$. Red, green and blue curves correspond to $\beta = 2, 5, 10$ respectively.

The non-recursive Chen's $\sqrt{t}$ approach does not allow to match any particular memory budget making a like-for-like comparison difficult. Instead of fixing the memory budge, it is possible to fix computational cost at 2 forwards iterations on average to match the cost of the $\sqrt{t}$ algorithm and observe how much memory would our approach use. Memory usage by the $\sqrt{t}$ algorithm would be equivalent to saving $\sqrt{t}$ hidden states and $\sqrt{t}$ internal core states. Lets suppose that the internal RNN core state is $\alpha$ times larger than hidden states. In this case the size of the internal RNN core state excluding the input hidden state is $\beta = \alpha - 1$. This would give a memory usage of Chen's algorithm as $\sqrt{t}(1 + \beta) = \sqrt{t}(\alpha)$, as it needs to remember $\sqrt{t}$ hidden states and $\sqrt{t}$ internal states where input hidden states can be omitted to avoid duplication. Figure 7 illustrates memory usage by our algorithm divided by $\sqrt{t}(1 + \beta)$ for a fixed execution speed of 2 as a function of sequence length and for different values of parameter $\beta$. Values lower than 1 indicate memory savings. As it is seen, we can save a significant amount of memory for the same computational cost.

Another experiment is to measure computational cost for a fixed memory consumption of $\sqrt{t}(1 + \beta)$. The results are shown in Figure 7. Computational cost of 2 corresponds to Chen's $\sqrt{t}$ algorithm. This illustrates that our approach does not perform significantly faster (although it does not do any worse). This is because Chen's $\sqrt{t}$ strategy is actually near optimal for this particular memory budget. Still, as seen from the previous paragraph, this memory budget is already in the regime of diminishing returns and further memory reductions are possible for almost the same computational cost.

## 5 Conclusion

In this paper, we proposed a novel approach for finding optimal backpropagation strategies for recurrent neural networks for a fixed user-defined memory budget. We have demonstrated that the most general of the algorithms is at least as good as many other used common heuristics. The main advantage of our approach is the ability to tightly fit to almost any user-specified memory constraints gaining maximal computational performance.

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
