[Supplementary Material]

# A  Pseudocode for BPTT-HSM

Below is a pseudocode of an algorithm which evaluates an optimal policy for BPTT-HSM. This algorithm has a complexity of $O(t^2 \cdot m)$ but it is possible to optimize it to $O(t \cdot m)$ by exploiting convexity in $t$. In any case, this is a one-off computation which does not have to be repeated while training an RNN.

---

**Algorithm 1:** BPTT-HSM strategy evaluation.

---

**input** : $t_{max}$ – maximum sequence length; $m_{max}$ – maximum memory capacity

1  EVALUATESTRATEGY($t_{max}, m_{max}$)
2      Let $C$ and $D$ each be a 2D array of size $t_{max} \times m_{max}$
3      **for** $r \in \{1, \ldots, t_{max}\}$ **do**
4          $C[t][1] \leftarrow \frac{t(t+1)}{2}$
5          **for** $m \in \{t, \ldots, m_{max}\}$ **do**
6              $C[t][m] \leftarrow 2t - 1$
7              $D[t][m] \leftarrow 1$
8      **for** $m \in \{2, \ldots, m_{max}\}$ **do**
9          **for** $t \in \{m + 1, \ldots, t_{max}\}$ **do**
10             $C_{min} \leftarrow \infty$
11             **for** $y \in \{1, \ldots, t - 1\}$ **do**
12                 $c \leftarrow y + C[y][m] + C[t - y][m - 1]$
13                 **if** $c < C_{min}$ **then**
14                     $C_{min} \leftarrow c$
15                     $D[t][m] \leftarrow y$
16             $C[t][m] \leftarrow C_{min}$
17     **return** $(C, D)$

---

Algorithm 2, shown below, contains pseudo-code which executes the precomputed policy.

---

**Algorithm 2:** BPTT-HSM strategy execution.

---

**input** : $D$ – previously evaluated policy; rnnCore – mutable RNN core network; stack – a stack containing memorized hidden states; gradHidden – a gradient with respect to the last hidden state; $m$ – memory capacity available in the stack; $t$ – subsequence length; $s$ – starting subsequence index;

1  EXECUTESTRATEGY($D$, rnnCore, stack, gradHidden, $m, t, s$)
2      hiddenState = PEEK(stack)
3      **if** $t = 0$ **then**
4          **return** gradHidden
5      **else if** $t = 1$ **then**
6          output $\leftarrow$ FORWARD(rnnCore, GETINPUT($s$), hiddenState)
7          gradOutput $\leftarrow$ SETOUTPUTANDGETGRADOUTPUT($s + t$, output)
8          (gradInput, gradHiddenPrevious) $\leftarrow$ BACKWARD(rnnCore, GETINPUT($s$), hiddenState,
9                                                              gradOuput, gradHidden)
10         SETGRADINPUT($s + t$, gradInput)
11         **return** gradHiddenPrevious
12     **else**
13         $y \leftarrow D[t][m]$
14         **for** $i \in \{0, \ldots, t - 1\}$ **do**
15             output, hiddenState $\leftarrow$ FORWARD(rnnCore, GETINPUT($s + i$), hiddenState)
16         PUSH(stack, hiddenState)
17         gradHiddenR $\leftarrow$ EXECUTESTRATEGY($D$, rnnCore, stack, gradHidden, $m - 1, t - y, s + y$)
18         POP(stack)
19         gradHiddenL $\leftarrow$ EXECUTESTRATEGY($D$, rnnCore, stack, gradHiddenR, $m, y, s$)
20         **return** gradHiddenL

---

# B Upper bound of the computational costs for BPTT-SHM

## B.1 General upper bound

Consider the following dynamic program:

$$C(t, m) = \min_{1 \le y < t} (y + C(t - y, m - 1) + C(y, m)) \tag{11}$$

with boundary conditions: $C(t, 1) = \frac{1}{2}t(t + 2)$ and $C(t, m) = 2t - 1$ for all $m \ge t$.

**Proposition 1.** *We have $C(t, m) \le mt^{1+1/m}$ for all $t, m \ge 1$.*

*Proof.* It is straightforward to check that the bound is satisfied at the boundaries. Now let us define the boolean functions $A(t, m) := \{C(t, m) \le mt^{1+1/m}\}$ and $A(m) := \{\forall t \ge 1, C(t, m) \le mt^{1+1/m}\}$.

We prove by induction on $m$ that $A(m)$ is true. We accomplish this by assuming that $A(m)$ is true and proving that $A(m + 1)$ is also true. We will first prove, by induction on $t$, that $A(t, m + 1)$ is true. Thus for any $t \ge 2$, assume $A(y, m + 1)$ is true for all $y < t$ and let us prove that $A(t, m + 1)$ is also true.

We have

$$\begin{aligned} C(t, m + 1) &= \min_{1 \le y < t} \left[ y + C(t - y, m) + C(y, m + 1) \right] \tag{12} \\ &= \min_{1 \le y < t} \left[ y + m(t - y)^{1+1/m} + (m + 1)y^{1+1/(m+1)} \right] \tag{13} \end{aligned}$$

using our inductive assumptions that both $A(y, m + 1)$ and $A(t - y, m)$ are true.

For any real number $y \in [1, t - 1]$, define $g(y) := y + m(t - y)^{1+1/m} + (m + 1)y^{1+1/(m+1)}$. Note that $g$ is convex (as the sum of 3 convex functions) and is smooth over $[1, t - 1]$ with first and second derivatives

$$g'(y) = 1 - (m + 1)(t - y)^{1/m} + (m + 2)y^{1/(m+1)},$$

and

$$g''(y) = \frac{m + 1}{m}(t - y)^{1/m - 1} + \frac{m + 2}{m + 1}y^{1/(m+1) - 1}.$$

Notice that $g''$ is positive and convex, thus

$$\max_{1 \le s \le t - 1} |g''(s)| = \max(g''(1), g''(t - 1)) \le \frac{m + 1}{m}(1 + (t - 1)^{1/m - 1}) \le 4. \tag{14}$$

Let $y^*$ be the (unique) optimum of $g$ (i.e., such that $g'(y^*) = 0$). Then we have

$$\begin{aligned} C(t, m + 1) &\le g(\lfloor y^* \rfloor) \\ &\le g(y^*) + (y^* - \lfloor y^* \rfloor)g'(y^*) + \frac{1}{2}(y^* - \lfloor y^* \rfloor)^2 \max_{1 \le s \le t - 1} |g''(s)| \\ &\le g(y^*) + 2 \\ &\le g(\tilde{y}) + 2 \end{aligned}$$

where $\tilde{y}$ defined as the unique solution to

$$t - y = y^{m/(m+1)}. \tag{15}$$

Notice that for any $t \ge 2$, $\tilde{y} \in [1, t)$. We deduce from (13) that

$$\begin{aligned} C(t, m + 1) &\le \tilde{y} + m(t - \tilde{y})^{1+1/m} + (m + 1)\tilde{y}^{1+1/(m+1)} + 2 \\ &\le \tilde{y} + m\tilde{y} + (m + 1)\tilde{y}^{1+1/(m+1)} + 2 \tag{16} \end{aligned}$$

Now, using the convexity of $x \mapsto x^{1+1/(m+1)}$ and since $y < t$, we have

$$
\begin{aligned}
t^{1+1/(m+1)} &\geq \tilde{y}^{1+1/(m+1)} + (t - \tilde{y})(1 + \frac{1}{m+1})\tilde{y}^{1/(m+1)} \\
&= \tilde{y}^{1+1/(m+1)} + \tilde{y}^{m/(m+1)}(1 + \frac{1}{m+1})\tilde{y} \\
&= \tilde{y}^{1+1/(m+1)} + (1 + \frac{1}{m+1})\tilde{y} \quad (17)
\end{aligned}
$$

(where the last equality derives from the definition of $\tilde{y}$ in (15)). Now putting (17) into (16) we deduce:

$$
\begin{aligned}
C(t, m+1) &\leq \tilde{y} + m\tilde{y} + (m+1)\big[t^{1+1/(m+1)} - (1 + \frac{1}{m+1})\tilde{y}\big] + 2 \\
&= (m+1)t^{1+1/(m+1)} - \tilde{y} + 2 \\
&\leq (m+1)t^{1+1/(m+1)},
\end{aligned}
$$

as soon as $\tilde{y} \geq 2$, which happens when $t \geq 4$. Finally, the cases $t < 4$ (which actually corresponds to the 2 cases: $(t = 3, m = 2)$ and $(t = 3, m = 3)$) are verified numerically. □

## B.2 Upper bound for short sequences

The algorithm described in the previous section finds an optimal solution using dynamic programming. It is trivial to show that $C(t, m)$ is an increasing function in $t$ and we will make use of this property. It is possible to derive a computational upper bound by finding a potentially sub-optimal policy and evaluating its cost. Alternatively, one can find a policy for a given computational cost, and then use this as an upper bound for an optimal policy.

Recall, from Section 3, that when the memory limit is the same as the sequence length ($t = m$), then $C(t, m) = 2t - 1 < 2t$. Define $T(a, m)$ as the maximum sequence length $t$ for which an average computational cost $C(t, m)/t \leq a$. We refer to the quantity $a$ as a *cost per time step* (CPTS). We can see that $T(2, m) \geq m$.

**Proposition 2.** $T(a, m) \geq \frac{m^{a-1}}{(a-1)!}$.

*Proof.* This is satisfied for the case $a = 2$. Assume that the proposition is true for some value $a$. We prove by induction that this is also satisfied for all other values of $a$.

Figure 8: Illustration of the optimal strategy for two easy cases: a) $t = 4, m = 4$ and b) $t = 10, m = 4$. $T(a, m)$ is the maximum sequence length for which $C(t, m) \leq at$. Note that we can construct more expensive fixed-CPTS strategies by composing cheaper fixed-CPTS strategies. Each sub-strategy in Figure b has one less memory slot available when going to the right. In the case of arbitrary sequence lengths, strategies may not look as nice.

Consider a sequence of length $t = \sum_{i=1}^{m} T(a, i)$. We will make the first initial forward pass over the sequence at the cost $t$ and will memorize hidden states spaced at intervals $T(a, m), T(a, m - 1)$

.. $T(a, 2)$ (Figure 8). Once the hidden states are memorized, we can run the same backpropagation algorithm on each sub-sequence each time paying the cost of $\leq a$ per time-step (Proposition 2). This will make the local cost of the algorithm $C(t, a+1) \leq t + ta = (a+1)t$. As this makes the cost per time-step $\leq a$, $T(a+1, m) \geq t = \sum_{i=1}^{m} T(a, i) \geq \sum_{i=1}^{m} \frac{i^{a-1}}{(a-1)!} \geq \frac{m^a}{a!}$.

$\square$

This implies that $C(t, m) \leq (a+1)t$ when $t \leq \frac{m^a}{a!}$. Note that this is a much better result than a strategy that sub-divides the interval into equal sub-intervals several times recursively, as the latter strategy would only give $C(t, m) \leq (a+1)t$ for $t \leq \frac{m^a}{a^a}$ while $(a! \ll a^a)$ for the same computational cost. Unfortunately, it is non-trivial to invert this function to express the computation explicitly as a function of $t$ and $m$.

### B.3   Analytical upper bounds for BPTT-ISM and BPTT-MSM

When memorizing internal core states instead of hidden states is allowed, we can apply almost exactly the same calculation of the upper bound as in Section 3.5 while still being conservative in our bound estimates. The main difference, though, is the removal of a single forward operation per time-step. This would give us the upper bound of the computational cost as $C(t, m) \leq at$ for $t \leq \frac{m^a}{a!}$. The same upper bound of $C(t, m) \leq mt^{1+\frac{1}{m}}$ can be assumed to be true for the case when $t > \frac{m^a}{a!}$, because an internal state also includes a hidden state, and the derived optimal algorithm can not do any worse for the same number $m$; however, even for the same $m$ more absolute memory would be required, as the units of measurement are different. $T(a, m)$ is the maximum sequence length for which a computational cost $C(T(a, m), m) \leq at$.

Any upper bounds derived for the case of BPTT-HSM will also hold for the case of BPTT-MSM, because the later is generalization of the former technique, and an optimal policy found will be at least as good as.

## C   Generalizing to deep feed-forward networks

BPTT-MSM can also be generalized to deep network architectures rather than RNNs, as long as the computational graph is a linear chain. The main difference is that different layers in deep architectures are non-homogeneous (*i.e.,* have different computational costs and memory requirements) while cores within the same RNN are homogeneous. This difference can be addressed by modifying the dynamic program formulation.

To start with, let us derive a strategy when only hidden states are stored. We can recall that an optimal policy of BPTT-HSM algorithm could be found by solving given dynamic programming equations.

If we choose to memorize next state at position $y$ and thereafter follow an optimal policy, the cost is:

$$Q(t, m, y) = y + C(t - y, m - 1) + C(y, m). \tag{18}$$

The optimal position of the next memorization is

$$D(t, m) = \operatorname*{argmin}_{1 \leq y < t} Q(t, m, y), \tag{19}$$

and computational cost under the optimal policy:

$$C(t, m) = Q(t, m, D(t, m)). \tag{20}$$

As in the case of RNNs, we choose to remember only some of the intermediate output results (hidden states) and recompute all internal states (and other hidden states) on demand while fitting within some memory limit. As sizes of internal representations of different layers are different, it is necessary to include the size of a "working" network layer into our current memory allowance. In the case of RNNs, this constant factor could be comfortably ignored. In addition, hidden states produced by different layers will also have different sizes.

Suppose that the cost of recomputing layer $y$ is $u_y$ while the size of a hidden state computed after step $y$ has the size of $s_y$. We also assume that the initial input vector is always available.

Define $U(x, y) := \sum_{i=x+1}^{x+y} u_i$ as a cumulative computational cost of forward propagation when $x$ bottom layers are cut-off and we are forward propagating over $y$ layers. Also, define $p_i$ to be the size of an internal state of some given network layer: specifically, as the minimum memory requirement to execute forward and backward operations on a given layer. Neither forward nor backward operations are possible if less memory is available than required by the operation. Define the maximum memory usage when executing forward operation on layers from $x + 1$ to $y$ inclusive as $P(x, y) = \max_{x < i \leq y} p_i$. For the reasons discussed previously, it is convenient to set the computational cost to infinity when we have have less than required memory available: $K(x, y, m) = 0$ if $P(x, y) \leq m$ and $K(x, y, m) = \infty$ if $P(x, y) > m$.

Consider the upper part of the neural network with $x$ bottom layers cut-off. Define $C(t, m - 1, x)$ as a computation cost of a combined forward and back-propagation on such network over $t$ bottom layers.

The cost of a combined forward and back-propagation of the cut-off section assuming that the next memorization happens at position $y$ is:

$$Q(t, m, y, x) = U(x, y) + K(x, y, m) + C(t - y, m - s_{x+y}, x + y) + C(y, m, x). \qquad (21)$$

$K(x, y, m)$ prevents us from making an impossible back-propagation commitment when memory is not enough.

It is now trivial to define position of the next memorization as:

$$D(t, m, x) = \underset{1 \leq y < t}{\operatorname{argmin}} \, Q(t, m, y, x), \qquad (22)$$

and can evaluate the cost under the optimal policy:

$$C(t, m, x) = Q(t, m, D(t, m, x), x). \qquad (23)$$

As previously, we have to set boundary conditions. When no extra memory is left:

$$C(t, 0, x) = \sum_{i=x}^{x+t} (t - i + 1) u_i, \qquad (24)$$

and when hen the memory is plentiful ($m >= \sum_{i=x}^{x+t} s_i$):

$$C(t, m, x) = u_t + \sum_{i=x}^{x+t-1} 2u_i. \qquad (25)$$

There is another boundary condition applies when the network has zero depth: $C(t, m, x) = 0$ when $x + t > N$ and $N$ is the number of layer in the network and also $C(t, m, x) = 0$ when $t \leq 0$. It is convenient to set $C(t, m, x) = \infty$ for $m < 0$ to emphasize that memory can never become negative.

We can solve the equations using dynamic programming, but unlike in the recurrent case it requires filling a 3D rather than a 2D array. This means that the evaluation of the strategy may become impractical for sequences longer than a few hundred time-steps, but fortunately only needs to be computed once.

The algorithm is executed as follows: if any any point we start a recursive call of the algorithm at layer $x$ while having memory allowance $m$, we evaluate $y = D(t, m, x)$, forward-propagate states until $y$, memorize the next state and call the algorithm recursively both parts of the sequence.

Equations can be generalized similarly for the BPTT-ISM and BPTT-MSM variants.