[Reviews · NeurIPS 2016]

Reviewer 1

Summary

This paper proposes an effective memory allocation strategy to save GPU memory consumption for back propagation through time algorithm. The basic idea is using dynamic programming to find an optimal memory usage policy to balances the tradeoff between memorization and recomputation. Empirical result shows that under a fixed computational cost for forward propagation the proposed method achieves about half the memory usage compared with a previous method.

Qualitative Assessment

Limited memory size of modern GPUs is a bottleneck to train very deep neural network. In order to alleviate this problem, this paper proposes an effective memory allocation strategy to save GPU memory consumption for back propagation through time algorithm. Theoretical analysis and bounds are well presented. The main drawbacks are the experiment part. I believe that if the paper could provide more evidence about its potential influence in real applications, it will be a very interesting paper. The detailed comments are as follows, -- There is few experiments on large-scale dataset or real applications. Without those empirical evidence, it is still not very clear how is the speedup or memory usage reduction of the proposed method comparing with other methods. -- The experiment result is not very significant since there is almost no speedup for very long sequences with a feasible memory consumption. -- The idea of dynamic programming adopted in this paper is a little incremental compared with Chen’s divide-and-conquer algorithm. -- Besides memory cost, there are other factors limit the efficient training of very deep neural networks, such as convergence speed and computational cost in each iteration. The help of saving a small part of memory consumption is very limited in training deep neural networks, unless the saving is significant. It will be very helpful if we could consume orders of magnitude less memory. -- Minor: In page 7, “as expensive than a orward operation” --> “as expensive than a forward operation”. Many typos in supplementary materials.

Confidence in this Review

1-Less confident (might not have understood significant parts)


Reviewer 2

Summary

Training recurrent neural networks requires unfolding the network in time and computing the backpropagation "through time". In order to perform it efficiently and to avoid recomputing intermediate values, these are usually stored in memory. However, as considered sequences increase in length and networks grow in size, the amount of available memory might not be sufficient. An usual solution is to truncate the backpropagation, resulting in approximate weight updates. It is also possible to recompute the forward pass, which is inefficient in computation time. Finally, heuristical "divide-and-conquer" approaches have been proposed to trade off between computation time and memory efficiency. The authors of this paper propose a new solution, in order to find the optimal strategy to get an algorithm both memory-efficient and not too expensive in terms of computation. More particularly, they define a cost as the number of forward operations needed to compute BPTT given a fixed memory budget. This cost is minimized with dynamic programing, in terms of when to store the intermediate results in memory versus when to recompute them. They study storing the hidden state of recurrent networks (i.e. output of the recurrent neurons), the internal state (needed to compute this output) or both, which each correspond to a different formulation of the cost. They show that this simple method can achieves 95% memory saving while being only 33% slower, and derive theoretical bounds in terms of the sequence length and memory budget. Their method is compared to naive baselines and to an handcrafted heuristical method.

Qualitative Assessment

This paper is very well written and pleasant to read. Although the figures supporting the explanations take some room, every bit is important and adds to the clarity of the paper. The idea to use dynamic programing to define when to store the intermediates in order to limit the number of forward operations in fixed memory budget scenarii is quite simple and well defined. The notation is clear and the figures are helpful. The chosen baselines are relevant (limit cases + heuristical divide-and-conquer method previously published). More importantly, this paper addresses a real problem, given the popularity of recurrent neural networks these days, along with the increasing size of models designed and sequences processed. I believe that in that scenario, many experimenters give up exact computation and use a truncated version of the backpropagation through time (quite a few papers in the last few years). This paper provides a simple and elegant solution, with all necessary details to implement it, including pseudo-code, which is highly appreciable. Following are a few remarks and cosmetic issues: - I did not quite understand how the 33% time increase can be deduced from Fig.5a. (end of p4) - Sect. 3.2 : typo in second paragraph ( "on less memory slot" ) - Fig 2b (in caption) : "per time step. sured in ms" - Sect. 4.2 : missing cap "in figure 7" (last paragraph) - Suppl., p3, last line : pb with ref: "Section ??" - Right-hand side of Fig.7 I guess the point was to show that curves overlap, but one can hardly see those curves. More generally, a legend on the figures rather than inside the captions would be appreciated, and increase the clarity of the figures. The titles and axes labels could also be bigger.

Confidence in this Review

2-Confident (read it all; understood it all reasonably well)


Reviewer 3

Summary

The paper proposes a method for performing back-propagation through time for training recurrent neural networks that can be accommodate computers with much less operating memory than required for the usual method of unfolding in time. The method computes the optimal policy for storing the internal state during network unfolding by means of dynamic programming. Multiple storage schemes are explored, where either the entire internal state is stored, or the activations of the hidden units only, or a mixture of both. Computational analysis demonstrates that the memory requirements can be reduced by 95% with a relatively minimal increase in computations (33% more). The method is likely to be very useful for training RNNs with long sequences on GPUs with limited memory.

Qualitative Assessment

I think this could potentially be a very useful paper that might expand significantly the length of sequences to which BPTT is applied. Some evidence that would reinforce this impression would be an actual motivating example, where a particular computing architecture (e.g., GPU model) runs out of memory with computing resources to spare. Otherwise, it is not entirely clear that memory would become the bottleneck for long sequences, and not the computing units. A minor typo: On page 7, in the third sentence of Section 4: "orward" -> "forward"

Confidence in this Review

3-Expert (read the paper in detail, know the area, quite certain of my opinion)


Reviewer 4

Summary

This paper proposed a memory-efficient backpropagation through time by using dynamic programming, besides they provided an analytical upper bound for their strategy. Comparing to Chen's algorithm, their proposed BPTT-MSM can fit within almost arbitrary constant memory constraints which is the main advantage.

Qualitative Assessment

As we known, memory consumption is a bottleneck in training complex recurrent neural networks on very long sequences. This paper proposed a memory-efficient backpropagation through time by using dynamic programming. However, compared to Chen's work, the strategies of this paper are more complex and Chen has tested on specific tasks using mxnet. I wonder if you could provide some results on specific rnn networks like seq2seq learning.

Confidence in this Review

1-Less confident (might not have understood significant parts)


Reviewer 5

Summary

The process of training recurrent neural network models can be quite hungry for memory. With sequences that aren't necessarily of fixed-length, the memory allocation can vary depending on the training sample. Traditionally, RNN/LSTM models have been trained with "full memoization" -- that is, during forward-propagation, all states through time are stored to be reused during backpropagation. If this is too expensive, "Chen's algorithm" uses less memory (but far more computation) by recomputing each hidden state during backpropagation. In this paper, the authors expose a "slider" in resource usage from "high computation / low memory" to "low computation / high memory." This slider works by choosing how many states to memoize (1 of every 3 states, spaced evenly ... or 1 out of 4 states, etc). I have indeed heard complaints from LSTM researchers about memory usage, and this seems like a good solution. So far as I can tell, the authors do not evaluate their results on any dataset, nor do they train a real RNN/LSTM model to convergence.

Qualitative Assessment

(Here, I use "RNN" as a shorthand for "RNN, LSTM, and related recurrent models.") The authors are solving an important problem. RNN training procedures can be greedy for memory. And, given the sequential nature, it's not trivial to simply to scale the training of each sequence over many machines. As a result, it's important to judiciously use memory and computational resources to train RNNs efficiently. I'm pleased to see the authors not only proposing a new instance of a solution, but to provide a user-selectable tradeoff between the quantity of computation and the memory usage. *Real Experiments.* I think the authors should really have an experimental result on a real dataset. - The short version: I can't give above a "2" rating for "experimental methods are appropriate" because... there aren't really any experiments! - The long version: RNN models have been applied to a variety of sequence-oriented domains, including automatic classification/understanding of text, audio, and video data. In most of these problem domains, many authors have released code for their RNN-based solutions. My "not-an-RNN-expert" colleague was able to download code, install it, and launch a training run of an RNN for UCF-101 video activity recognition over a two-day period. With all of that in mind.... I *really* think the authors should use their method to train an actual RNNon an actual dataset and to report results! With the same RNN architecture, the same training protocol, and the same random seed: I *think* the results (trained model) should be numerically equivalent to "traditional" RNN training with full memoization. And, the authors would be able to state actual numbers for memory savings and increased computational requirements. - If the authors can run such an experiment and deliver sane results, I would give a higher score for Technical Quality. *Motivation.* The paper would be stronger if you'd say, for example, "In [some application], we've seen that an RNN needs 1000GB of memory, and this just isn't feasible! Here's now we save >100x memory with a modest increase in computational footprint."

Confidence in this Review

2-Confident (read it all; understood it all reasonably well)


Reviewer 6

Summary

This paper studies efficient memory usage for training RNNs. A dynamic programming solution is proposed for optimal memory allocation when training RNNs, allowing RNN training to handle long sequences with limited memory and at relatively low computational cost.

Qualitative Assessment

The dynamic programming formulation is a solid step forward on top of Chen et al’s fixed square root of t as well as the fixed recursive schedule. Being able to formulate the allocation problem as a dynamic programming problem also makes it clear that the optimal allocation can be found efficiently. Overall I think this is a nice and solid contribution. The distinction between hidden states and internal states is a little confusing. The results of this two cases also only differ by a constant factor, I wonder if it will be clearer to move some of the material to the appendix. On the other hand, I found the theoretical analysis in the appendix to be interesting. The only thing that is more obviously missing is the application of the proposed methods on real problems. It will be good to see if the ability to train on longer sequences help improve RNN performance, and see if there are any practical issues that occur beyond the theoretical analysis. A few minor problems: - This paper used the wrong format, it should use the template for submissions - At the beginning of section 3, it is said to discuss two cases, when memory is scarce and when memory is somewhat limited. However the second case was not discussed, or I somehow missed it? The authors never made the distinction between “scarce” and “somewhat limited” clear.

Confidence in this Review

2-Confident (read it all; understood it all reasonably well)